



# Sensitivity of urban boundary layer dynamics to surface characteristics of built terrains

Jiyun Song[1], Zhi-Hua Wang[1]

[1]School of Sustainable Engineering and the Built Environment, Arizona State University, Tempe, AZ 85287, USA

*Correspondence to*: Zhi-Hua Wang (zhwang@asu.edu)

**Abstract.** Urban land–atmosphere interactions can be captured by numerical modeling framework by coupling the land surface processes and atmospheric dynamics, while the model performance depends largely on accurate input parameters. In this study, we use an advanced stochastic approach to quantify parameter uncertainty and model sensitivity of a coupled numerical framework for urban land-atmosphere interactions. It is found that the development of urban boundary layer is

highly sensitive to surface characteristics of built terrains. In addition, hydrothermal properties of conventional and green roofs have different impacts on atmospheric dynamics due to different surface energy partitioning mechanisms. As far as atmospheric physical dynamics in the convective boundary layer is concerned, the fraction of paved/vegetated terrains imposes more significant impact than the urban morphology. The sensitivity analysis deepens our insight into the fundamental physics of urban land–atmosphere interactions and provides useful guidance for urban planning under

challenges of changing climate and urban expansions.

## 1 Introduction

Land surface connects soil layers and the overlying atmosphere by transferring momentum, heat, and water through the interface. Thus landscape characteristics are critical in determining surface heat and moisture fluxes, which in turn regulates the atmospheric boundary layer dynamics in, e.g. mesoscale atmospheric modeling (McCumber and Pielke, 1981). Despite

significant improvements of climate model predictability made in last decades, significant uncertainty still exists in model structures (i.e. mechanisms and equations), model parameters, numerical stability consideration, and scale transition (e.g. downscaling) (Hargreaves, 2010; Maslin and Austin, 2012). Statistical analyses on observational and numerical datasets have shown that land–atmosphere interaction is an importance source of uncertainty in climate predictability (Betts et al., 1996; Orlowsky and Seneviratne, 2010; Trier et al., 2011). Land–atmosphere interactions have significant impacts on

climate both temporally (from seasonal to interannual) and spatially (from local to global) (Seneviratne and Stöckli, 2008). The predictive skill and robustness of regional and global climate models can be significantly improved with a better representation of coupled land-atmospheric processes, specifically the soil moisture/temperature/precipitation interactions (Chen and Avissar, 1994; Chen and Dudhia, 2001; Phillips and Klein, 2014; Seneviratne et al., 2010).



Numerical weather and climate model uncertainties are further complicated due to the presence of complex built terrains. With a relatively small areal coverage, urban areas are manifested as hotspots of modified hydrothermal properties, altered flow fields, high surface heterogeneity, and anthropogenic heat and moisture sources (Arnfield, 2003; Flagg and Taylor, 2011; Wang et al., 2011b). Through land-atmosphere coupling, urban areas further impact hydroclimate in regional and even

global scales via modified surface energy and water cycles. Thus sensitivity analysis is critical to quantify model uncertainties and improve model predictability, as the model performance is largely dependent on the accuracy of input parameters. With prescribed atmospheric forcing (i.e. air temperature, pressure, humidity, wind speed, and solar radiation) such as by measurements in the surface layer, the convective boundary layer (CBL) dynamics are largely dictated by boundary characteristics at the bottom and top of the CBL. In particular, previous studies have found that critical parameters

for urban land surface modeling include urban morphology, roof properties, and characteristics of the inversion layer (Loridan et al., 2010; Wang et al., 2011a; Wong et al., 2011; Ouwersloot and Vilà-Guerau de Arellano, 2013).

The conventional approach to analyze model sensitivity is to change only one parameter with all the other parameters fixed and compare the output results with the "control case" (i.e. results from original unchanged parameter sets). This approach, however, will result in high computational costs with large sets of parameters and potentially biased statistical correlations

between uncertain parameters. On the other hand, statistical approaches handling the complete set of parameter uncertainty simultaneously in one simulation, e.g. those using Monte Carlo methods, are more suitable than the conventional sensitivity analysis (Wang et al., 2011a). For complex numerical framework involving multiple physics and large number of uncertain parameters, however, "curse-of-dimensionality" may arises in direct Monte Carlo simulations (MCS), a phenomenon that an algorithm works in low dimensions can break down in high dimensions (Bellman and Rand, 1957; Cherchi and Guevara,

2012). The curse-of-dimensionality necessitates more advanced Monte Carlo procedure, using importance sampling technique to improve computational efficiency. In this study, we will adopt the Subset Simulation (Au and Beck, 2001) based on Markov-Chain Monte Carlo (MCMC) procedure for sensitivity analysis, for urban land-atmosphere interactions using a one-dimensional (1D) numerical framework by coupling an urban land surface model with a single column atmospheric model (Song and Wang, 2015a). Subset Simulation is especially efficient for handling large dimensions of

uncertain space and simulating small probability events (climate extremes, for example) with either short- or long-tail behavior, which has been widely used in many engineering problems (Au et al., 2007; Thunnissen et al., 2007).

The paper is organized as follows. Physical parameterization schemes of the coupled SLUCM-SCM framework and procedures of Subset Simulation will be introduced in Section 2. In Section 3, three critical model responses (i.e. CBL height, virtual potential temperature and specific humidity of the mixed layer) will be evaluated; sensitive quantification of

uncertain model input parameters will be presented; and statistical error for Subset Simulation will also be assessed. More detailed discussions on sensitivity results with practical implications of urban planning on the CBL physical dynamics will be given in Section 4, followed by our concluding remarks in Section 5.





## 2 Methodology

### 2.1 Coupled urban land-atmospheric model

In this paper, urban land-atmosphere interactions are modeled using a 1D stand-alone and scalable numerical framework (Song and Wang, 2015a), by coupling an advanced single layer urban canopy model (SLUCM) for urban land surface

processes (Wang et al., 2011b; Wang et al., 2013) and a single column model (SCM) for boundary-layer dynamics (Noh et al., 2003; Troen and Mahrt, 1986). The schematic of the coupled SLUCM-SCM framework is shown in Fig. 1, which captures three vertical layers. The lowest level is the surface layer, which is considered as the constant flux layer and consists of 10% of the entire CBL with the built terrain located at the bottom. The middle level is a convective mixed layer, where distributions of temperature and humidity are determined by buoyant plumes arising from the surface layer and atmospheric

turbulence. The top level is an entrainment zone with a temperature inversion, which inhibits upward mixing, and confines subjacent air and pollution in the CBL. Temperature and humidity profiles in the entire vertical column are regulated by heat and moisture fluxes exchanged across the interfaces of two adjacent layers.

At the bottom of numerical framework, the urban canopy layer is parameterized by a SLUCM, which is also adopted in the latest version of Weather Research and Forecast (WRF) model (v3.7.1) (Yang et al., 2015). This new SLUCM features

enhanced urban hydrological processes coupled with the urban energy balance model, which enables a more realistic representation of the transport of energy and water over built terrains. The energy balance equation for the urban canopy layer is given by:

$$R_n + A_F = H_u + LE_u + G_0 , \tag{1}$$

where $R_n$ is the net radiation; $A_F$ is the anthropogenic heat fluxes; $H_u$ and $LE_u$ are the turbulent sensible and latent heat fluxes

arising from the entire urban canopy layer respectively; $G_0$ is the conductive heat flux aggregated over urban sub-facets (i.e. roof, wall, and ground), where the actual thickness and thermal mass of these solid media have been taken into account.

The turbulent sensible and latent heat fluxes arising from the urban area ($H_u$ and $LE_u$) are the areal average of those from roofs ($H_R$ and $LE_R$) and the street canyon ($H_{can}$ and $LE_{can}$) (Wang et al., 2013),

$$H_u = r \sum_{k=1}^{N_R} f_{R,k} H_{R,k} + w H_{can} , \tag{2}$$

$$LE_u = r \sum_{k=1}^{N_R} f_{R,k} LE_{R,k} + w LE_{can} , \tag{3}$$

while the turbulent fluxes from street canyon are aggregated over walls and the ground,

$$H_{can} = \frac{2h}{w} \sum_{k=1}^{N_W} f_{W,k} H_{W,k} + \sum_{k=1}^{N_G} f_{G,k} H_{G,k} , \tag{4}$$

$$LE_{can} = \frac{2h}{w} \sum_{k=1}^{N_W} f_{W,k} LE_{W,k} + \sum_{k=1}^{N_G} f_{G,k} LE_{G,k} , \tag{5}$$



where $N_R$, $N_W$ and $N_G$ are the number of sub-facet types of roofs, walls and ground (road) respectively; $f_R$, $f_W$ and $f_G$ are the fraction of sub-facet types of roofs, walls and ground respectively; $r = R/(R+W)$, $w = W/(R+W)$, and $h = H/(R+W)$ are the normalized roof width, canyon width, and building height, respectively, with $R$, $W$, and $H$ the physical dimensions. By assuming that surface layer is a constant-flux layer, the turbulent fluxes at the top of surface layer are the same with those arising from the urban canopy ($H_u$ and $LE_u$).

To resolve the overlying atmospheric boundary layer, a modified version of the Yonsei University (YSU) boundary layer scheme commonly used in the WRF model (Hong et al., 2006; Noh et al., 2003) was applied by incorporating an analytical prognostic formula (Ouwersloot and Vilà-Guerau de Arellano, 2013) rather than a diagnostic formula related with Richardson number (Hong et al., 2006) for determining the boundary layer height. In the mixed layer, the governing equation for mean profiles of virtual potential temperature and specific humidity due to boundary layer turbulence in SCM is given by (Troen and Mahrt, 1986):

$$\frac{\partial \theta_v}{\partial t} = \frac{\partial}{\partial z}(-\overline{w'\theta_v'}) , \tag{6}$$

$$\frac{\partial q}{\partial t} = \frac{\partial}{\partial z}(-\overline{w'q'}) , \tag{7}$$

where $\theta_v$ is the virtual potential temperature; $q$ is the specific humidity; $w$ is the vertical wind speed; and $\overline{w'X'}$ with $X = \theta_v$ or $q$ is the vertical kinematic eddy flux, with the over-bar denoting the ensemble average. The vertical kinematic eddy heat and moisture flux at the lower-boundary of the mixed layer is given by

$$\left(\overline{w'\theta'}\right)_s = \frac{H_s}{\rho_a c_p} , \tag{8}$$

$$\left(\overline{w'q'}\right)_s = \frac{LE_s}{\rho_a L_v} , \tag{9}$$

where subscript $s$ denotes the atmospheric surface layer; $\theta$ is the potential temperature; $\rho_a$ is the density of the air; $c_p$ is the specific heat of air at constant pressure; and $L_v$ is the latent heat of vaporization of water. From the definition of virtual potential temperature, we have

$$\left(\overline{w'\theta_v'}\right)_s = 0.61\overline{\theta}\left(\overline{w'q'}\right)_s + \left(1+0.61\overline{q}\right)\left(\overline{w'\theta'}\right)_s . \tag{10}$$

The upper boundary condition is at the height of CBL ($z_h$), which appears as a mixing height scale in turbulence closure schemes in climate and weather prediction models and acts as an impenetrable lid for pollutants released at the surface (Zilitinkevich and Baklanov, 2002). Here, we use an analytical method proposed by (Ouwersloot and Vilà-Guerau de Arellano, 2013) to determine $z_h$:

$$z_h = \left\{ z_{h0}^2 + \frac{(2+4w_e)}{\gamma_{\theta v}}\left[\Delta\theta_{v,0}z_{h0}^{\frac{1+w_e}{w_e}} - \left(\frac{w_e}{1+2w_e}\right)\gamma_{\theta v}z_{h0}^{\frac{1+2w_e}{w_e}}\right]\left(\hat{z}_h^{-\frac{1}{w_e}} - z_{h0}^{-\frac{1}{w_e}}\right) + \left(\frac{2+4w_e}{\gamma_{\theta v}}\right)\int_{t_0}^{t}\left(\overline{w'\theta_v'}\right)_s dt \right\}^{1/2} , \tag{11}$$





where $z_{h0}$ is the initial CBL height, $w_e$ is the entrainment rate at the inversion, $\gamma_{\theta v}$ is the lapse rate in the free atmosphere, $\Delta \theta_s$ is the potential temperature difference across the inversion, and $\hat{z}_h$ is a correction term given by

$$\hat{z}_h = \left[ z_{h0}^2 + \left( \frac{2+4w_e}{\gamma_{\theta v}} \right) \int_{t_0}^{t} \left( \overline{w'\theta_v'} \right)_s dt \right]^{1/2} . \tag{12}$$

The turbulent kinematic heat and moisture fluxes at the upper boundary of mixed layer (Hong et al., 2006; Kim et al., 2006) are

$$\left( \overline{w'\theta_v'} \right)_{z_h} = -0.15 \left( \frac{\theta_v}{g} \right) w_m^3 / z_h , \tag{13}$$

$$\left( \overline{w'q'} \right)_{z_h} \approx 0 , \tag{14}$$

where $w_m$ is the velocity scale for entrainment ($w_m^3 = w_*^3 + 5u_*^3$), which can be derived from the mixed layer velocity scale $w_*$ and surface friction velocity scale $u_*$; and $w_*$ is parametrized by

$$w_* = \left[ \frac{g}{\theta} \left( \overline{w'\theta'} \right)_s z_h \right]^{1/3} , \tag{15}$$

accounting for the surface heat flux $\left( \overline{w'\theta'} \right)_s$ at lower boundary of mixed layer and CBL height $z_h$. Equation (13) implies that the entrainment heat flux is closely related to the surface layer states. In large eddy simulations, the heat flux at the entrainment/inversion is usually estimated by

$$\frac{\left( \overline{w'\theta_v'} \right)_{z_h}}{\left( \overline{w'\theta_v'} \right)_s} = -A_R , \tag{16}$$

with $A_R = 0.15$ typically (Hong et al., 2006; Kim et al., 2006; Noh et al., 2003). The upward heat flux from land surface and the downward heat flux at the inversion layer both enhance turbulent mixing in the mixed layer.

The kinematic turbulent heat and moisture flux in the mixed layer with the account of non-local mixing and entrainment effect can be parameterized as (Noh et al., 2003).

$$-\overline{w'\theta_v'} = K_h \left( \frac{\partial \theta_v}{\partial z} - \gamma_h \right) - \left( \overline{w'\theta_v'} \right)_h \left( \frac{z}{z_h} \right)^3 , \tag{17}$$

$$-\overline{w'q'} = K_h \left( \frac{\partial q}{\partial z} - \gamma_q \right) - \left( \overline{w'q'} \right)_h \left( \frac{z}{z_h} \right)^3 , \tag{18}$$

where $K_h$ is turbulent diffusivity which is assumed to be identical for heat and moisture transport; $z$ is the vertical distance from surface; $\gamma_h$ and $\gamma_q$ are non-local mixing terms (Noh et al., 2003; Troen and Mahrt, 1986), given by



$$\gamma_h = C \frac{\left(\overline{w'\theta'}\right)_s}{w_s z_h}, \tag{19}$$

$$\gamma_q = C \frac{\left(\overline{w'q'}\right)_s}{w_s z_h}, \tag{20}$$

where $C$ is a coefficient of proportionality, often set as 6.5 according to Troen and Mahrt (1986) and $w_s$ is the velocity scale for the entire CBL. With prescribed initial states (i.e. profiles of $\theta_v$ and $q$) and boundary conditions given by Eqs. (8)-(14), we can readily estimate the time evolution and vertical profiles of temperature and humidity in the CBL based on the above physical parameterization schemes.

## 2.2 Model evaluation

To evaluate the coupled SLUCM-SCM framework outlined in Section 2.1, experiment data of temperature and humidity profiles were obtained from NOAA/ESRL radiosonde database (http://esrl.noaa.gov/raobs/) for two typical convective days, i.e. July 2$^{nd}$, 2013 and July 9$^{th}$, 2013 at Phoenix site (33.45 N, 111.95 W), Arizona. All atmospheric data in the ESRL Radiosonde database were subjected to gross error and hydrostatic consistency checks according to Schwartz and Govett (1992). The coupled modeling framework was driven by meteorological data measured in the closest wireless meteorological station (33.44 N, 111.92 W) to be included in the footprint area of the radiosonde site. The comparison of the simulated and observed profiles of virtual potential temperature and specific humidity is shown in Fig. 2 for the two days at 16:44 pm and 16:37 pm (local time), respectively. The good agreement between predicted and measured temperature and humidity profiles in the boundary layer not only validates the parameterization of CBL dynamics, but also the accuracy of SLUCM predictions, particularly the total sensible and latent heat arising from the urban area. Note that the integrated SLUCM-SCM framework can be readily tested on the WRF platform in an online setting, i.e. by coupling with other dynamic modules (e.g. radiation, Noah land surface model for natural terrains, etc.). Here we focused on the sensitivity of the offline (stand-alone) SLUCM-SCM framework to exclude the physical and numerical perturbation (e.g. model stability) that could potential arouse from the online testing with coupling to mesoscale dynamics (e.g. regional advection, synoptic influence, etc.).

## 2.3 Subset Simulation

In urban climate modeling, the capability of assessing critical responses of atmospheric processes to urban land use land cover change is of paramount significance for assessment of climatic extremes. The SLUCM-SCM framework coupling urban land surface processes and CBL dynamics involves a large number of input parameters, which leads to high dimensionality of input space for the following statistical analysis. Hence we adopt Subset Simulation (Au and Beck, 2001) for subsequent sensitivity study, which is efficient in simulating rare (very small probability) events and robust for high dimensionality. Instead of simulating rare events as in direct MCS method with expensive computational cost, Subset Simulation breaks down extreme events with small exceedance probability into a sequence of more frequent events by





introducing intermediate exceedance events. The targeted small exceedance probability is then expressed as a product of larger conditional probabilities of each intermediate event. In addition, MCMC technique is adopted based on effective accept/reject rules in Subset Simulations to improve computational efficiency.

As illustrated in Fig. 3, the sampling technique employed in the Subset Simulation proceeds as follows: In level 0 (initial state), the unconditional samples of uncertain parameters follow a prescribed probability distribution function (PDF) (Fig. 3a). Conditional samples in level 1 are defined using a given intermediate conditional probability $p_0$ (e.g. $p_0 = 0.1$ stands for 10% of the level 0 samples will be selected as conditional samples) (Fig. 3b). These samples are then generated by MCMC procedure using importance sampling at the exceedance probability $P(Y > y_1) = p_0$ ($Y$ is a critical response of model and $y_1$ is a threshold value) (Fig. 3c). Subsequent conditional sampling are conducted by MCMC with the intermediate exceedance probability target, i.e. $P(Y > y_i) = p_0^i$ ($i = 1, 2, 3, \ldots$ denoting conditional levels) until simulations reach the final target with $p_f = p_0^N$, where $p_f$ is the target probability of a rare event and $N$ the total number of conditional levels (Fig. 3d). Using this method, a rare event, e.g. with target exceedance probability of $p_f = 10^{-4}$ (i.e. the probability of occurrence is less than 1 in 10,000), can be effectively broken down into 4 different sampling (1 unconditional MCS and 3 subsequent conditional MCMC) levels, each samples a moderate conditional probability of $p_0 = 0.1$.

## 3 Results of sensitivity analysis

In this section, we apply Subset Simulation to analyze the sensitivity of the coupled SLUCM-SCM to different input parameters. The meteorological forcing in the surface layer was prescribed using field measurements of an eddy covariance tower on a clear day (14 June 2012) provided by the Central Arizona-Phoenix Long Term Ecological Research (CAP LTER) project (Chow et al., 2014). The inputs of diurnal air temperature, relative humidity, and downwelling shortwave and longwave radiation are plotted in Fig. 4, with the daytime from 6:00 am to 7:30 pm (local time) for the development of CBL. With the prescribed meteorological forcing, the surface sensible and latent heat fluxes are predicted by the SLUCM, which then in turn drive the SCM to estimate temperature and humidity profiles in the mixed layer. The calibrated input parameters of SLUCM-SCM (including surface dimensional and hydrothermal parameters for the SLUCM and atmospheric parameters for the SCM) are presented in Table 1. Note that the initial soil water content for green roofs in the SLUCM is set as 90% saturated for the subsequent 13.5-hour of simulation after the beginning of CBL development such that the evaporative power of green roofs is not constrained by soil water availability. Among the model inputs, 15 parameters were selected (see Table 2) for uncertainty evaluation based on previous related studies (Ouwersloot and Vilà-Guerau de Arellano, 2013; Wang et al., 2011a; Yang and Wang, 2014b). Since the initial parameter distribution by direct MCS are pivotal to the statistical sampling efficiency of Subset Simulations, PDFs for uncertain parameters are carefully selected to constitute a physically realistic parameter space. In addition, it was found that normal (Gaussian) distribution is more realistic for thermal and hydrological parameters with the expected value in a physical range having higher probability, while the distributions of dimensional parameters are subject to engineering design and is therefore more uniform (Wang et al., 2011a). The two



atmospheric parameters at the top of CBL (i.e. entrainment rate and lapse rate) are also set as uniform distribution to achieve same probability for different top boundary conditions according to Ouwersloot and Vilà-Guerau de Arellano (2013).

### 3.1 Critical model responses

Three atmospheric variables, i.e. the critical CBL height ($z_h$), the mean virtual potential temperature ($\theta_v$), and the mean
specific humidity ($q$) in the mixed layer are selected as model responses to assess the impact of urban land surface characteristics on the overlying atmosphere. By critical, it means that extreme responses of these model outputs (with small exceedance probability, or equivalently as "climatic extremes") are simulated using MCMC procedure. This is particularly relevant when urban planning is concerned with mitigation strategies of extreme events associated with future land use and climatic changes. For each monitored output, we simulate three different cases with the fraction of green roof vegetation of
0, 0.5, and 1.0, respectively. Note that we do not include vegetation on ground (though the model is capable of), so roof vegetation is the only moisture source. This model set-up allows us to analyze exclusively the effectiveness of green roofs, one of the urban environmental mitigation strategies of particular interest to researchers and city planners. For all three cases, three conditional levels are used with a conditional probability of $p_0 = 0.1$, which is equivalent to a sequence of exceedance probabilities of $10^{-1}$, $10^{-2}$, and $10^{-3}$ for MCMC levels 1, 2 and 3, respectively. In total, 270 simulations were run (30
independent simulations per case for 9 cases) with 1450 realizations of the set of 15 uncertain parameters in each run to ensure the simulation results are statistically significant.

Plots of exceedance probabilities versus various model responses averaged over 30 simulations are presented in Fig. 5. The variations of critical model outputs with three different green roof fractions indicate the sensitivity of roof greening degrees on CBL dynamics. In Fig. 5(a)&(b), we monitored CBL height and virtual potential temperature of mixed layer under three
conditions of green roof fractions (i.e. $f_{veg} = 0$, 0.5, and 1). In general, larger green roof fractions lead to lower $z_h$ and smaller $\theta_v$. This is expected since urban landscapes with larger fraction of vegetation distribute solar energy into more latent heat and less sensible heat, due to evaporative cooling. Less sensible heat and reduced surface temperature both lead to reduced CBL height and virtual potential temperature.

It is also noteworthy that there exist log concavities for the exceedance probabilities of both critical $z_h$ and $\theta_v$ with $f_{veg} = 1$
(100% roof greening). The occurrence of log concavities is related to energy balance in the street canyon where nonlinear effect of canyon aspect ratio $h/w$ was observed (Song and Wang, 2015a). Detailed explanations of aspect ratio effects will be described in Section 4.1. In Fig. 5(c), we monitored specific humidity of mixed layer under three conditions of green roof fractions (i.e. $f_{veg} = 0.1$, 0.5, and 1). As roof is set as the only moisture source, urban land surface is completely dry with $f_{veg} = 0$ and resulted in no moisture in the atmosphere in the absence of horizontal advection. Larger green roof fraction tends to
produce higher $q$ in the overlying CBL. In contrast to $z_h$ and $\theta_v$, exceedance probability distribution of critical response of $q$ does not exhibit log concavity because the moisture source is purely from roofs and canyon aspect ratio and building density have no contribution.



### 3.2 Statistical quantification of model sensitivity

In general, for an uncertain parameter, the deviation between the distribution of MCMC-generated conditional samples (in levels 1, 2, and 3) and the initial prescribed distribution sampled using direct MCS (level 0) indicates the significance of parameter sensitivity with respect to the corresponding model output. Figure 6 shows the comparison between conditional

distribution (histograms) and initial distribution (dashed line) for two sample parameters, i.e. heat capacity of green roof $C_{Rv}$ and canyon aspect ratio $h/w$ respectively, for a typical simulation with $f_{veg} = 1.0$ and critical $q$ as model output. It is clear that the critical response of $q$ is more sensitive to $C_{Rv}$ with noticeable deviation of sample distribution at each conditional level (Fig. 6a), while $h/w$ is relatively insignificant in influencing $q$ with small deviation of sample distribution (Fig. 6b). The result is physical as variation of $C_{Rv}$ affects roof surface energy balance, which in turn influences the humidity profile in the

CBL through surface moisture flux. On the contrary, since green roofs are the only moisture source in our setting, altering $h/w$ has negligible effect on the atmospheric moisture for the street canyon with no vegetation on ground or wall.

To better quantify the parameter sensitivity, a percentage sensitivity index (PSI) (Wang et al., 2011a) is adopted here to measure the model sensitivity to an uncertain parameter $X$ by calculating the average deviation of conditional sample means to that of the original PDF:

$$\text{PSI}[X] = \frac{1}{N} \sum_{i=1}^{N} \frac{E[X \mid Y > y_i] - E[X]}{E[X]}, \tag{21}$$

where $i$ is the conditional (MCMC) level index, $N = 3$ the total conditional levels, $E[X]$ the statistical mean (expected value) of the original unconditional distribution in level 0 (as in Table 2), $E[X/Y > y_i]$ the mean value of $X$ at conditional level $i$, $Y$ the value of monitored model response, and $y_i$ the threshold values at exceedance probability of each intermediate level $i$. The magnitude of PSI quantifies the significance of sensitivity, while the sign of PSI indicates the correlation between

monitored output $Y$ and input parameter $X$, i.e. positive PSI means increasing $X$ will lead to an increase of output $Y$ and negative PSI means increasing $X$ will lead to a decreased $Y$.

PSI values of all uncertain parameters for three different monitored outputs, i.e. $z_h$, $\theta_v$, and $q$, with different green roof fractions are presented in Table 3. For better visualization, bar plots of PSI values are also shown in Fig. 7. As shown in Fig. 7(a) and (b), both $z_h$ and $\theta_v$ are highly sensitive to surface dimensional parameters, including normalized roof width $r$,

canyon aspect ratio $h/w$, and roughness length of momentum for conventional roofs $Z_{m,Rc}$. Note that $r$ is positively correlated with critical $z_h$ and $\theta_v$ for conventional roofs while the correlation is negative for green roofs. Both critical $z_h$ and $\theta_v$ are negatively correlated with $h/w$ and positively correlated with $Z_{m,Rc}$. Moderate sensitivity of critical $z_h$ and $\theta_v$ is found with respect to thermal parameters of conventional roofs including albedo $a_{Rc}$, heat capacity $C_{Rc}$, and thermal conductivity $k_{Rc}$. Also note that there are opposite correlations for atmospheric parameters $w_e$ and $\gamma_{\theta v}$: $z_h$ is positively correlated with $w_e$ and

negatively correlated with $\gamma_{\theta v}$; but the correlations are opposite for model output of critical $\theta_v$. From Fig. 7c, mixed layer $q$ is highly sensitive to $r$ and thermal properties of green roofs and moderately sensitive to $Z_{m,Rv}$. Physical mechanisms governing the model sensitivity and its implications to urban planning are discussed below.





## 4 Discussion

Unprecedented rate of rapid urban expansion in last few decades has led to numerous environmental problems such as urban heat island (UHI) effect, degradation of air quality, and increase of building energy consumption (Arnfield, 2003). In particular, the UHI effect has attracted significant effort, even heated debate from urban climate researchers and city
planners. UHI is characterized by elevated temperature in built environments compared to surrounding rural areas (Oke, 1982). Major contributors of UHI include: (a) excess storage of thermal energy due to radiative trapping by street canyon and thermal properties of pavement materials, (b) reduced vegetation cover and evaporative cooling and (c) the release of anthropogenic heat, moisture, and greenhouse gases (Santamouris, 2014; Sun et al., 2013). Correspondingly, there are several popular UHI mitigation strategies, including (1) changing canyon geometry (characterized by aspect ratio and
roughness lengths) to alter the energy distribution through radiative shading and trapping; (2) changing thermal properties, such as installing cool roofs or cool pavements to reflect more solar radiation by increasing surface albedo; (3) adding green spaces, such as green roofs to increase evapotranspiration in urban area. We will discuss the effects of these UHI mitigation strategies on the overlying atmosphere based on the sensitivity study, and its implication to urban planning.

### 4.1 Impact of urban morphology

Building geometry and density in an urban area have a significant impact on the partitioning and redistribution of solar energy in the surface layer, which in turn modulate the energy transport processes in the overlying atmosphere. The canyon aspect ratio $h/w$ is a typical indicator of building geometry and density in urban planning (Ali-Toudert and Mayer, 2006; Krüger et al., 2011; Theeuwes et al., 2013). Low $h/w$ signals low building (small $h$) or sparse building density (large $w$), while high $h/w$ indicates high building (large $h$) or intensive building density (small $w$). With variable aspect ratio ranging
from 0.25 to 8, log concavity is found in the exceedance probability estimates for critical $z_h$ and $\theta_v$ in the case of $f_{veg} = 1.0$ as shown in Fig. 5(a) and (b). This log concavity is correlated with the nonlinear effect of the canyon aspect ratio on CBL height and virtual potential temperature, due to two counteracting processes, viz. shading effect and radiative trapping effect in the street canyon, as investigated by (Song and Wang, 2015a). To further test the nonlinear effect of $h/w$ on CBL dynamics, we set the canyon aspect ratio constant, and the log concavity disappears as shown in Fig. 8. The log concavity of
variable $h/w$ demarks the switching from small $h/w$ case to high $h/w$ case with a nonlinear interaction between radiative shading and trapping effects. In addition, at mesoscale atmospheric modeling, the canyon aspect ratio is closely related to the surface roughness of a built terrain, which in turn modulates the surface aerodynamic resistance under convective condition and further complicate the nonlinear effect.

### 4.2 Impact of thermal properties

As shown in Fig. 7, CBL states ($z_h$, $\theta_v$, and $q$) are moderately sensitive to surface thermal properties. Specifically, $a_{Rc}$, $C_{Rc}$, and $k_{Rc}$ of conventional roofs are important parameters in modulating $z_h$ and $\theta_v$, whereas $q$ is sensitive to $C_{Rv}$ and $k_{Rv}$ of green





roofs. Higher albedo causes more solar energy being reflected and less sensible heat arising from roofs, leading to smaller $z_h$ and $\theta_v$. Moderate model sensitivity to $a_{Rc}$ demonstrates that implementation of white/cool roofs with higher reflectivity is an effective way in reducing not only environmental temperature in the urban surface layer, but also the one in the overlying mixed layer.

It is also noteworthy in Fig. 7 that thermal properties of conventional roofs and those of green roofs have opposite correlation to different CBL dynamics, which can be explained by plausible mechanisms governing surface energy balance. For a conventional roof, larger heat capacity implies that more thermal energy is needed to heat the roof, while higher thermal conductivity implies that less time is needed for heat dissipation, both leading to lower roof surface temperature (Wang et al., 2011b). Lower roof surface temperature will then reduce the sensible heat (given other conditions invariant),
causing lower CBL height and lower temperature in the mixed layer, as shown in Fig. 7(a)&(b). In Fig. 7(c), it is shown that to increase $q$, more latent heat from green roofs needs to be supplied so that sensible heat will decrease. This potentially causes green roof surfaces to be cooler than the atmosphere, giving rise to the "oasis" effect commonly observed over surfaces with significant evaporative cooling (Stull, 1988). As a result, sensible heat flux can be negative and flowing towards the surface. Under this condition, larger heat capacity and thermal conductivity of green roofs increase the ground
heat flux, and are positively correlated to $q$ via evaporative cooling. Nevertheless, we emphasize here that what the PSI values can reveal is as good as that the coupled SLUCM-SCM framework can capture. The actual physics of urban land-atmosphere interactions involves more complicated land surface and atmospheric processes of heat and water transport in the integrated soil-atmosphere system due to complexity of surface energy partitioning (Yang and Wang, 2014a). For example, the existence of phase lags among land surface temperatures and energy budgets, due to subsurface heat transport with pore
water advection, can lead to complex hysteresis loops (Sun et al., 2013; Wang, 2014) that are not adequately captured by the current numerical framework.

### 4.3 Impact of green roofs

Due to their ability to modify energy and water budgets in the urban surface layer, city planners are increasingly using green roofs as an effective strategy to mitigate UHI effect (Sailor et al., 2012; Susca et al., 2011; Wang et al., 2016). In our study,
four sets of green roof parameters are studied: (1) thermal parameters, i.e. $a_{Rv}$, $C_{Rv}$, and $k_{Rv}$; (2) hydrological parameters, i.e. saturated soil water content $W_s$, residual soil water content $W_r$, and saturated hydraulic conductivity $K_s$; (3) roof width $r$; and (4) green roof fraction $f_{veg}$. Humidity in the CBL is moderately sensitive to green roof thermal properties with a positive correlation, as discussed above. In addition, all hydrological parameters are relatively insensitive as shown in Fig. 7. This is plausibly due to the initial soil moisture condition (90% saturated), which is realistic provided green roofs are carefully
maintained with constant irrigation. The assumption is also relevant in this study for more "manageable" urban surface characteristics for urban planning purpose. More detailed boundary layer physics and sensitivity related to soil water and hydrological properties of other urban vegetation (such as urban lawns, urban agriculture, etc.), on the other hand, require further investigation (Cuenca et al., 1996; Song and Wang, 2015b).





In contrast, CBL dynamics are very sensitive to green roof width and areal fractions, as they determine the area of green roof in a built environment, which in turn strongly influence the soil water availability for evaporation. It is shown that larger green roof width $r$ and fraction $f_{veg}$ lead to lower $z_h$, smaller $\theta_v$, and higher $q$ in the mixed layer as a result of evaporative cooling by green roofs. This result is expected and clearly indicates the effectiveness of green roofs in regulating

atmospheric dynamics above an urban area. To further test the effectiveness of green roofs, we monitored the same set of model outputs, viz. $z_h$, $\theta_v$ and $q$, with $f_{veg}$ ranging from 0% to 100% with an increment of 10%. Threshold values at three conditional sampling levels are plotted in Fig. 9, i.e. $y_i$ for $i = 1$, 2, and 3, with corresponding exceedance probability of $10^{-1}$, $10^{-2}$, and $10^{-3}$, respectively. For all output variables at different conditional levels, the results can be well fitted using linear relations with high $R^2$ values: $z_h$ and $\theta_v$ decrease linearly with the green roof fraction, while $q$ increases linearly with $f_{veg}$. As

far as UHI mitigation is concerned, the mean mixed layer temperature can be reduced by 3-4 K in either a more probable (level 1) or a more extreme (level 3) case with an increase of green roof fraction from 0 to 100%. It is noteworthy that in this study, the supply of soil water content to green roof systems is assumed to be ample (e.g. via urban irrigation). In an arid environment such as Phoenix, especially during drought, the trade-off between water (for irrigation) and energy (cooling load) needs to be carefully measured by city planners.

**4.4 Impact of roughness lengths**

Roughness lengths of momentum and heat transfer are important land surface characteristics that regulate the aerodynamic resistance related to turbulent transport of mass, momentum and energy in the surface layer (Grimmond and Oke, 1999). In this study, we set the roughness lengths of momentum at the roof level as uncertain parameters for both conventional and green roofs. The roughness lengths of heat transfer follow a simple parameterization that $Z_h = Z_m/10$. From Fig. 7, both $z_h$

and $\theta_v$ in the mixed layer are highly sensitive to $Z_{m,Rc}$, while $Z_{m,Rv}$ of green roofs plays an important role in regulating $q$. As indicated in Table 3, when critical $z_h$ is monitored, PSI value of $Z_{m,Rc}$ is 38.53% for $f_{veg} = 0$ and 34.42% for $f_{veg} = 0.5$; for critical $\theta_v$, PSI of $Z_{m,Rc}$ is 42.58% for $f_{veg} = 0$ and 24.38% for $f_{veg} = 0.5$. These high PSI values indicate a strong correlation between aerodynamic resistance of turbulent transfer and the CBL dynamics. This implies that altering roughness lengths of roofs is an effective way to influence energy transport from surface to the overlying CBL without fundamental changes to

the urban morphology or geometry in the street canyon.

In addition to urban landscape characteristics, the coupled SLUCM-SCM numerical framework also involves physical parameterizations at the top of CBL, i.e. in the inversion layer. The uncertainties of two atmospheric parameters, namely the entrainment rate $w_e$ and the lapse rate of virtual potential temperature $\gamma_{\theta v}$ are tested. From Fig. 7(a), $z_h$ increases with $w_e$ and decreases with $\gamma_{\theta v}$, as expected according to Eq. (11). From Fig. 7(b), impacts of $w_e$ and $\gamma_{\theta v}$ on critical mixed layer $\theta_v$ are

opposite. This is because larger $w_e$ or smaller $\gamma_{\theta v}$ result in larger $z_h$ according to Eq. (11), which further cause smaller non-local mixing effects according to Eqs. (19) and (20), leading to decrease of $\theta_v$ in the mixed layer.





Lastly, we evaluate the statistical quality of Subset Simulation by computing the coefficient of variation (c.o.v., defined as the ratio of the standard deviation to the mean) using a typical statistical average of 30 independent runs. The resulted c.o.v. of Subset Simulation as a function of exceedance probability is shown in Fig. 10, where c.o.v. of direct MCS is also shown for comparison. Estimate of c.o.v. of direct MCS can be analytically formulated as $[(1−p_i)/(p_i N_T)]^{1/2}$ (Au and Beck, 2001), where $p_i$ is the exceedance probability and $N_T$ the number of samples at corresponding MCMC level $i$. It is clear that the c.o.v. of Subset Simulation is significantly smaller than that of direct MCS, especially at the higher MCMC level (smaller probability), indicating less statistical error for exceedance probability estimates using Subset Simulation.

## 5 Concluding remarks

In this study, we use an advanced Monte Carlo method to quantify the sensitivity of atmospheric dynamics to urban land surface characteristics for a coupled urban land–atmosphere model. Results show that in general CBL dynamics over a built terrain are largely dictated by the urban morphology, roughness lengths, and hydrothermal properties of landscape materials. Specifically, thermal properties of conventional and green roofs exhibit different impacts on CBL height, mixed layer temperature, and humidity, due to different surface energy partitioning. For conventional roofs, increasing surface albedo (e.g. implementation of white roofs) is shown to be an effective way in reducing urban environmental temperature. On the other hand, the deployment of green roofs in an urban area seems to be a more effective urban planning strategy to mitigate adverse environmental problems, not only in reducing environmental temperature through evaporative cooling but also in better management of urban water cycles and overlying boundary layer moisture profiles. It is also intriguing to note that urban morphology, in this study represented as the street canyon aspect ratio, introduces nonlinear impact on the CBL height and mixed layer temperature, governed by interactions of two counteracting mechanisms of solar energy distribution in building arrays, viz. the radiative trapping effect in enhancing surface warming and the shading effect for surface cooling. This nonlinear interaction of surface energy processes is manifested in the dynamics of overlying atmospheric boundary layer. In the context of UHI mitigation, cool and green roofs are probably the two most popular methods that have received extensive research effort. Besides, changing roughness lengths or thermal properties on rooftops (e.g. by planting different species of vegetation for green roofs, or using porous pavement materials for conventional roofs) can also be effective means in reducing urban environmental temperatures (in the surface layer and CBL).

In addition, we would like to reiterate here that results of sensitivity analysis in this study are based on the model physics of the stand-alone coupled SLUCM-SCM numerical framework; the actual urban land-atmosphere interactions involve more complicated physical processes in transferring momentum, heat, and moisture in the soil-land-atmosphere continuum. Nevertheless, as various research groups worldwide have extensively tested the numerical framework, either separately or in integrated platforms (e.g. WRF), we are confident that this physically-based model captures the basic physics of urban land-atmosphere interactions. Results of sensitivity study of the numerical framework thus shed new light on the impact of urban



land surface characteristics on the overlying atmosphere, and provide useful guidelines for urban planning under future expansion and emergent climatic patterns.

## Acknowledgements

This work is supported by the National Science Foundation (NSF) under grant No. CBET-1435881. We thank Ms. Melissa Wagner for helping in obtaining Radiosonde data in Phoenix.

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




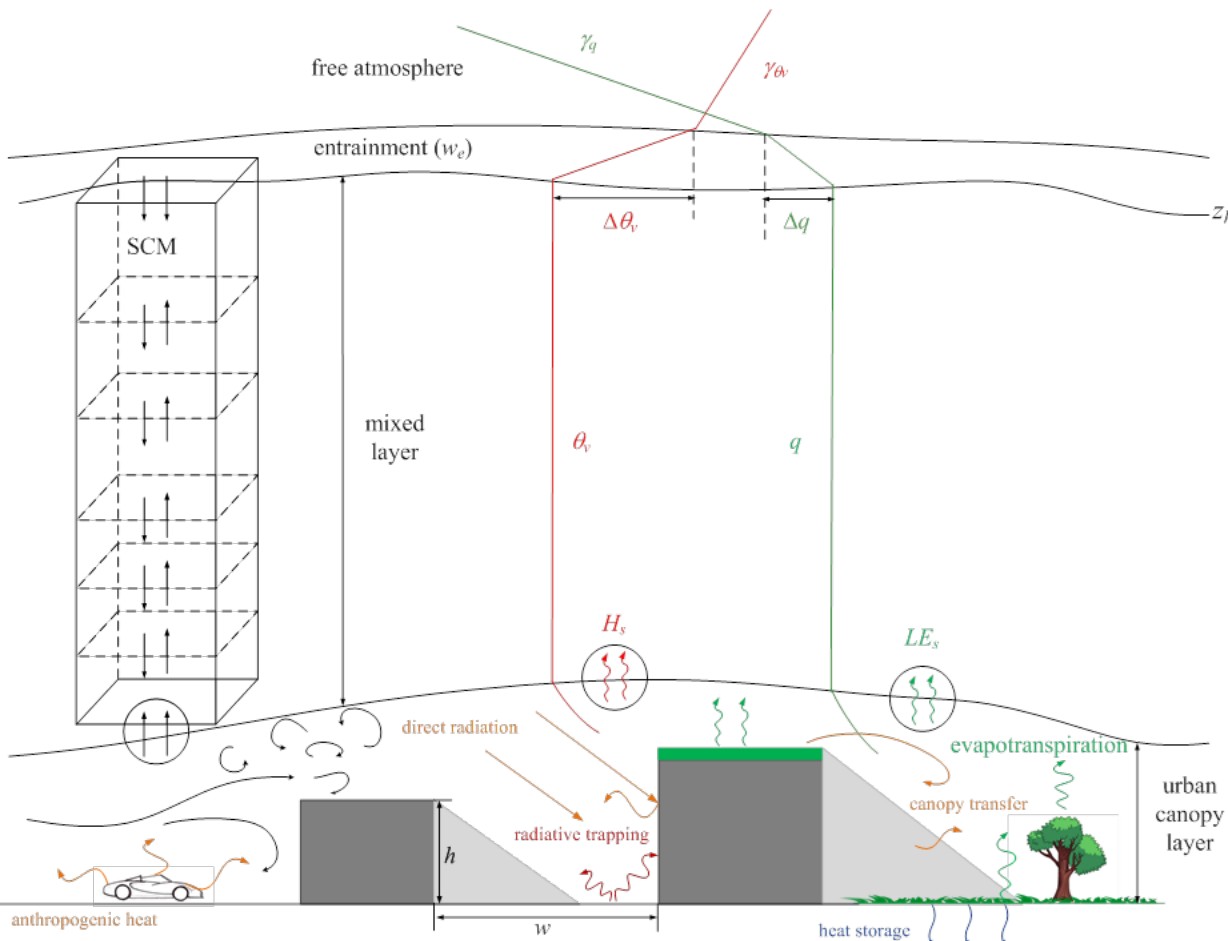

**Figure 1: Schematics of coupled SLUCM-SCM framework: land surface processes are parameterized by a single layer urban canopy model; atmospheric processes under convective condition are parameterized by a single column model.**





**(a)**

**(b)**

**Figure 2: Comparison of simulated and measured atmospheric profiles of virtual potential temperature** $\theta_v$ **and specific humidity** $q$ **for two time points, i.e. (a) 16:44 pm (local time) on July 2$^{nd}$, 2013, and (b) 16:37 pm (local time) on July 9$^{th}$, 2013 at NOAA-ESRL Phoenix site.**





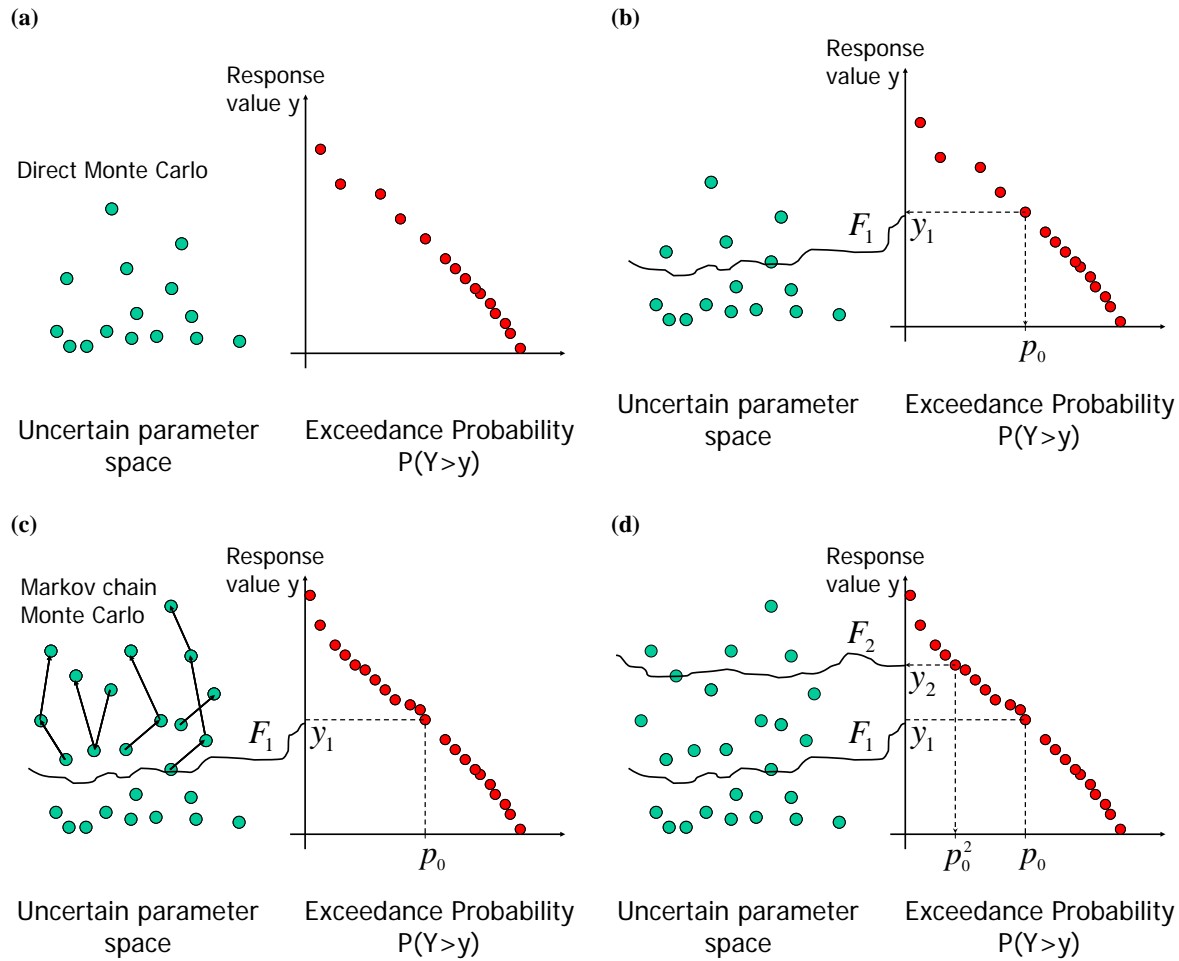

Figure 3: Schematic of Subset Simulation procedure: (a) level 0 (initial phase) sampling by direct MCS, (b) determination of level 1 samples $F_1$ given conditional exceedance probability $p_0$, (c) populating conditional samples in level 1 by MCMC procedure, and (d) forwarding algorithm to subsequent conditional levels till the target exceedance probability $p_f = p_0^N$ is reached.





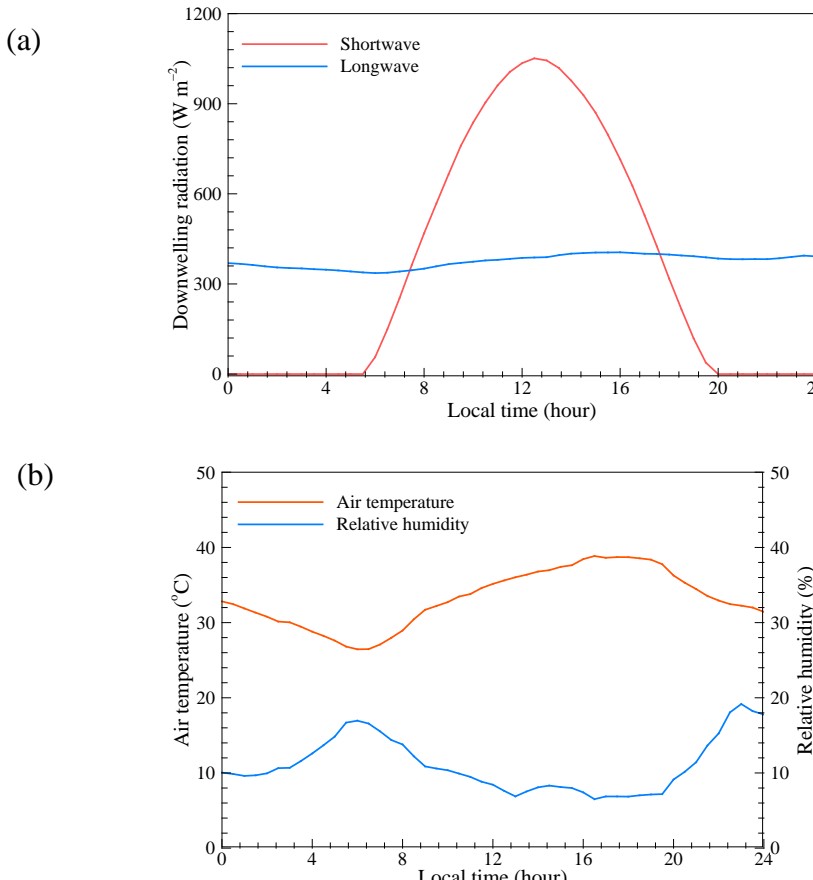

**Figure 4: The diurnal surface atmospheric forcing of June 14, 2012 (a clear day) in Phoenix, AZ: (a) downwelling shortwave and longwave radiation and (b) air temperature and relative humidity. The daytime data between starting point (6:00 am local time) and ending point (7:30 pm local time) are used to drive the SLUCM-SCM under convective condition.**





**Figure 5: Estimates of exceedance probabilities for model outputs of critical (a) CBL height, (b) virtual potential temperature, and (c) specific humidity with different green roof fractions.**





**Figure 6: Histogram of conditional samples at different conditional levels for (a) a sensitive parameter, and (b) an insensitive parameter for a typical simulation with $f_{veg} = 1.0$ and critical $q$ as model output.**





**Figure 7: PSI values for model outputs of critical (a) $z_h$, (b) mixed layer $\theta_v$, and (c) mixed layer $q$, with different green roof fractions.**




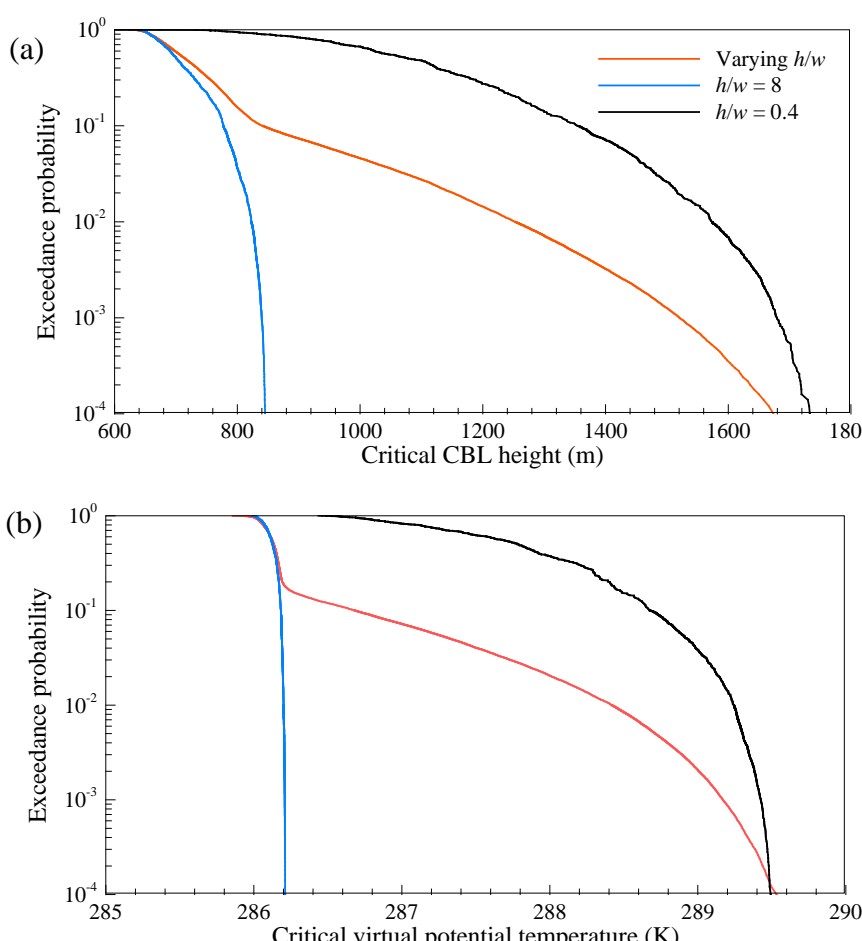

**Figure 8: Illustration of the nonlinear effect of apect ratio $h/w$ on critical model responses of (a) $z_h$ and (b) $\theta_v$ of the CBL.**





**Figure 9: Threshold values at different conditional levels as functions of green roof fractions for critical (a) $z_h$, (b) mixed layer $\theta_v$, and (c) mixed layer $q$. MCMC levels 1, 2 and 3 correspond to exceedance probabilities of $10^{-1}$, $10^{-2}$, and $10^{-3}$, respectively.**





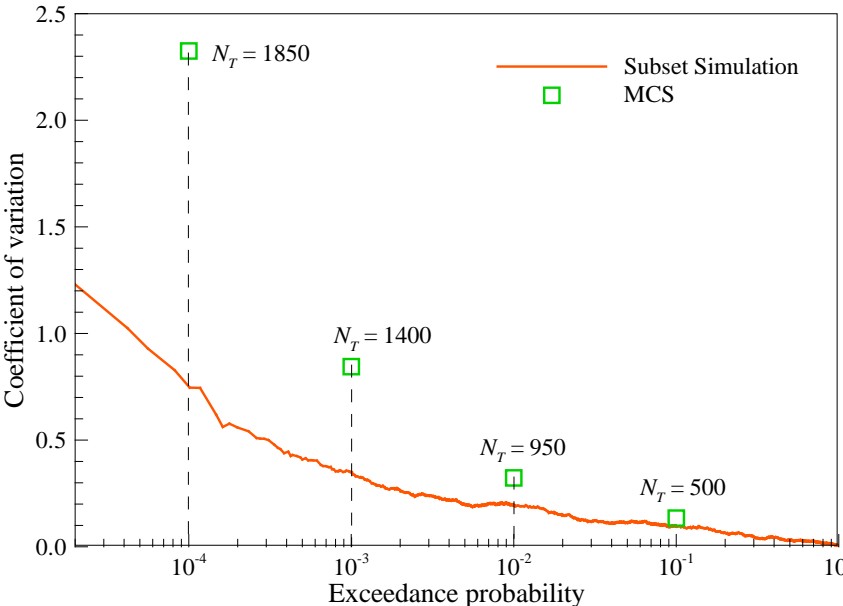

**Figure 10: Comparison of the coefficient of variation (c.o.v.) of exceedance probability in Subset Simulation and direct MCS.**





**Table 1: Input parameters of the coupled SLUCM-SCM numerical framework.**

| Input parameters | Symbol |
|---|---|
| *Surface dimensional parameters* | |
| Roof level (building height) (m) | $Z_R$ |
| Reference height of atmospheric measurements (m) | $Z_a$ |
| Aspect ratio (-) | $h/w$ |
| Roughness length for momentum above conventional roof (m) | $Z_{m,Rc}$ |
| Roughness length for heat above conventional roof (m) | $Z_{h,Rc}$ |
| Roughness length for momentum above vegetated roof (m) | $Z_{m,Rv}$ |
| Roughness length for heat above vegetated roof (m) | $Z_{h,Rv}$ |
| Roughness length for momentum above canyon (m) | $Z_{m,can}$ |
| Roughness length for heat above canyon (m) | $Z_{h,can}$ |
| *Surface thermal parameters* | |
| Albedo of conventional roof surface (-) | $a_{Rc}$ |
| Albedo of vegetated roof surface (-) | $a_{Rv}$ |
| Albedo of wall surface (-) | $a_W$ |
| Albedo of ground surface (-) | $a_G$ |
| Emissivity of conventional roof surface (-) | $\varepsilon_{R,c}$ |
| Emissivity of vegetated roof surface (-) | $\varepsilon_{R,v}$ |
| Emissivity of wall surface (-) | $\varepsilon_W$ |
| Emissivity of ground surface (-) | $\varepsilon_G$ |
| Thermal conductivity of conventional roof (W m$^{-1}$ K$^{-1}$) | $k_{R,c}$ |
| Thermal conductivity of vegetated roof (W m$^{-1}$ K$^{-1}$) | $k_{R,v}$ |
| Thermal conductivity of wall (W m$^{-1}$ K$^{-1}$) | $k_W$ |
| Thermal conductivity of ground (W m$^{-1}$ K$^{-1}$) | $k_G$ |
| Heat capacity of conventional roof (J m$^{-3}$ K$^{-1}$) | $C_{R,c}$ |
| Heat capacity of vegetated roof (J m$^{-3}$ K$^{-1}$) | $C_{R,v}$ |
| Heat capacity of wall (J m$^{-3}$ K$^{-1}$) | $C_W$ |
| Heat capacity of ground (J m$^{-3}$ K$^{-1}$) | $C_G$ |
| *Surface hydrological parameters* | |
| Saturated soil water content (soil porosity) (-) | $W_s$ |
| Residual soil water content (-) | $W_r$ |
| Saturated hydraulic conductivity (m s$^{-1}$) | $K_s$ |
| *Atmospheric parameters* | |
| Entrainment rate at the inversion (m s$^{-1}$) | $w_e$ |
| lapse rate of virtual potential temperature in the free atmosphere (K s$^{-1}$) | $\gamma_{\theta v}$ |



**Table 2: Summary of statistics of uncertain parameters used in the sensitivity study.**

| Type | Parameter | Unit | PDF | Min | Max | Mean | Std dev |
|---|---|---|---|---|---|---|---|
| Surface thermal parameters | $a_{Rv}$ | - | Normal | 0.05 | 0.6 | 0.18 | 0.045 |
| | $C_{Rv}$ | MJ m$^{-3}$ K$^{-1}$ | Normal | 0.1 | 2 | 0.72 | 0.18 |
| | $k_{Rv}$ | W m$^{-1}$ K$^{-1}$ | Normal | 0.15 | 4 | 0.85 | 0.213 |
| | $a_{Rc}$ | - | Normal | 0 | 1 | 0.15 | 0.0375 |
| | $C_{Rc}$ | MJ m$^{-3}$ K$^{-1}$ | Normal | 0.1 | 4 | 1.52 | 0.38 |
| | $k_{Rc}$ | W m$^{-1}$ K$^{-1}$ | Normal | 0.2 | 3 | 1.2 | 0.3 |
| Surface hydrological parameters | $W_s$ | - | Normal | 0.3 | 0.6 | 0.44 | 0.074 |
| | $W_r$ | - | Normal | 0.04 | 0.2 | 0.074 | 0.025 |
| | $K_s$ | m s$^{-1}$ | Normal | 0.1 | 100 | 1.7 | 0.43 |
| Surface dimensional parameters | $r$ | - | Uniform | 0.3 | 0.8 | - | - |
| | $h/w$ | - | Uniform | 0.25 | 8 | - | - |
| | $Z_{m,Rc}$ | mm | Uniform | 0.1 | 5 | - | - |
| | $Z_{m,Rv}$ | mm | Uniform | 10 | 200 | - | - |
| Atmospheric parameters | $w_e$ | m s$^{-1}$ | Uniform | 0.1 | 0.3 | - | - |
| | $\gamma_{\theta v}$ | K km$^{-1}$ | Uniform | 3 | 7 | - | - |





**Table 3: Estimates of PSI values for critical CBL height $z_h$, virtual potential temperature $\theta_v$, and specific humidity $q$ in the mixed layer, each averaged over 30 runs.**

| Uncertain parameters | $z_h$ | | | $\theta_v$ | | | $q$ | | |
|---|---|---|---|---|---|---|---|---|---|
| | $f_{veg} = 0$ | 0.5 | 0 | $f_{veg} = 0$ | 0.5 | 1 | $f_{veg} = 0.1$ | 0.5 | 1 |
| $a_{Rv}$ | -1.44 | -0.86 | 2.67 | -0.19 | 0.03 | 1.65 | 2.95 | 7.64 | 6.81 |
| $C_{Rv}$ | -0.46 | 3.12 | 0.21 | -0.59 | 0.27 | 0.03 | 22.23 | 37.78 | 39.76 |
| $k_{Rv}$ | 0.54 | -1.60 | -0.17 | 1.13 | -0.40 | 0.06 | 24.35 | 36.02 | 35.48 |
| $a_{Rc}$ | -15.67 | -8.30 | -1.60 | -18.69 | -11.79 | -0.37 | 3.73 | 6.09 | -0.98 |
| $C_{Rc}$ | -12.10 | -6.41 | -0.39 | -14.49 | -9.50 | -0.29 | 5.93 | 2.26 | -2.02 |
| $k_{Rc}$ | -17.60 | -11.29 | 0.67 | -23.33 | -11.35 | 0.65 | 2.89 | 3.93 | -0.91 |
| $W_s$ | 0.35 | -1.36 | 0.24 | -0.04 | 0.34 | -0.17 | -0.49 | 1.39 | 1.84 |
| $W_r$ | 1.63 | 1.41 | 2.70 | 0.44 | 0.22 | 2.26 | 3.59 | 1.11 | 1.00 |
| $K_s$ | -0.17 | -2.48 | 0.64 | 0.24 | -1.59 | -1.89 | -0.90 | 0.53 | -1.94 |
| $r$ | 29.73 | 5.59 | -25.08 | 34.60 | 2.65 | -26.72 | 27.95 | 33.55 | 37.17 |
| $h/w$ | -33.23 | -69.41 | -87.32 | -40.08 | -79.89 | -89.15 | 3.95 | 7.55 | 5.86 |
| $Z_{m,Rc}$ | 38.53 | 34.42 | 1.00 | 42.58 | 24.38 | -1.13 | -9.76 | -7.75 | 7.72 |
| $Z_{m,Rv}$ | 0.26 | -0.31 | -5.36 | 0.53 | -1.00 | -0.26 | -15.41 | -16.77 | -1.28 |
| $w_e$ | 20.16 | 19.78 | 10.91 | -14.61 | -14.93 | -6.65 | -1.45 | -4.03 | -1.55 |
| $\gamma_{\theta v}$ | -33.46 | -32.33 | -19.91 | 27.22 | 22.62 | 12.84 | 6.20 | 5.20 | 6.12 |