# Peer review of "Evaluating the impact of built environment characteristics on urban boundary layer dynamics using an advanced stochastic approach"

_Atmospheric Chemistry and Physics, 2015_

## Referee Comment (RC1) · Anonymous Referee #2 · 11 Mar 2016

The paper has incorporated reviewer's comments and addressed various issues/concerns raised by the reviewer. Firstly, the scientific values of the paper extend beyond the novel method presented here. There are many different land surface models coupled to the atmosphere (not just within the urban context) with large number of input parameters To access the sensitivity of high dimensional input parameter space, it is often a formidable task. Therefore, the application of subset simulation based on Markov Chain Monte Carlo in the context of evaluating urban land surface models or other numerical models will be promising. Secondly, sensitivity tests that take advantage of MCMC of handling small probability events can be potentially related to quantifying risks associated with these events. i.e. risks in urban environment

due to extreme events, such as a heat wave. This opens the possibility of combining atmosphere-coupled urban land surface model to risk assessment, which will potentially be useful in fully assessing the impact of climate extremes. Thirdly, the implications from the results of sensitivity test of this paper will be significant. One reason is that as mentioned in the paper, which is to inform the urban planners in terms of building green roofs. Another reason is that the results from sensitivity test shed light into better parameterizations of the parameters used in urban land surface models. Since land surface models often rely on parameterizations of turbulent momentum, heat and vapor fluxes, the high sensitivity of the convective boundary layer dynamics to turbulence parameterization in the urban land surface model provides further motivation to derive better representations of turbulence. Overall, the novel approach and results presented here will impact a broad range of audience. Therefore, I recommend acceptance of this paper.

---

## Referee Comment (RC2) · Anonymous Referee #1 · 21 Mar 2016

An advanced stochastic approach is presented to evaluate the sensitivity of a numerical model framework with offline setup. While the numerical approach is promising and could advance knowledge on the implications of set model parameters, the conclusions drawn about the model parameters are mostly not new.

General comments:

- This is not the first time model parameters are evaluated with such a stochastic approach. More insights should be provided on existing literature on this area of research. The open questions and uncertainties should be listed and addressed specifically. It should be clearly stated why certain parameters are being tested.

- The conclusions drawn on the sets of model parameters are mostly to be expected and often directly obvious from the equations. The analysis should be rephrased so that these findings are not presented as if they were new results. Rather the benefits of the stochastic approach compared to other techniques should be emphasized. The conclusions sections should be rephrased accordingly.

- It would also be appropriate to change the title to put the emphasize on the methodology rather than the parameter testing as this is the more relevant/new contribution of this work.

specific comments:

P1, l18: Be careful with such statements: 'fraction of paved/vegetated terrains imposes more significant impact than the urban morphology'. When comparing different drivers for conditions in the atmospheric boundary layer, it is difficult to draw such generalized conclusions. In this study, extremely high fractions of irrigated green roofs are tested which will naturally have a big impact. It would have to be evaluated how much the morphology would have to change to achieve similar effects in order to make such a direct, generalized comparison.

P1, l27: Make clear this is referring to the model setup: 'coupled' land-atmosphere processes?

P2, l9: should 'boundary characteristics' rather be 'boundary conditions'?

P2, l12: State which studies in the literature have used similar approaches to simulate the impact of varying input parameters, i.e. changing multiple parameters simultaneously. Which are the open research questions this study is aiming to address?

P2, l15: define 'uncertain parameters'

P2, l19: check grammar

P2, l29: explain what is meant by 'sensitive quantification'

P3, l19: comment on advection, i.e. that it is not represented by the used setup

P3, l21: what about the heat storage in the air volume within the street canyons?

P4, l4: where is the 'top of the surface layer'? mention blending height concept

P4, l25: repetition with line 8

P6, l12: Which database do the surface station data come from? How did you estimate the footprint of the radiosonde?

P6, l16: To draw the conclusion that turbulent surface fluxes are also evaluated by the good agreement with the radiosonde profiles includes a series of assumptions. Also comment on the differences between observations and model profiles in the lowest layers where surface fluxes will have the most impact.

P7, l22: What exactly is meant by 'calibrated' input parameters? Are all roofs at mean roof height or is some variability assigned to the roof height?

P7, l24: Why did you choose such high soil moisture levels?

P7, l27: Comment why these specific parameters were selected/ why the others are considered less critical.

P7, l30: How did you select physically realistic parameter space for uncertain parameters? Based on literature? Representing a specific type of city?

P7, l32: By 'dimensional parameters' you mean morphometric?

P8, l1: How did you set uniform distribution for entrainment rate and lapse rate?

P8, l24: with 'log concavities' you are referring to the shape of the curves? Maybe this could be re-phrased to me something more descriptive.

P9, l23: As Table 3 and Figure 7 have the same content, the table could be moved to supplementary material?

[Figure]

P9 l24: Normalised roof width not included in Table 1. Is a change in roof width connected to a change in plan area fraction?

P10, l2-13: move to introduction

P10, L23: According to Table 1, aerodynamic resistance appears to be a set input parameter. Here you mention it to be a function of surface roughness. If this is the case this should be mentioned in the methods section.

P11, l29: Careful: most green roofs are constructed to be extensive, i.e. that they would not require irrigation but would use only natural water from precipitation. Comment on the representativeness of your model setup.

P11, l32: this sentence is difficult to read. What is meant by 'detailed boundary layer physics . . . require further investigation'?

P12, l10: Are these results integrated over the whole day?

P12, l19: Provide reference for the Zm – Zh relation

P12, l23: Given resistances are directly related to turbulent fluxes, this is not a surprising result. Altering the roughness length is a fundamental change to urban morphology. Especially as roughness characteristics of individual facets combine to the local scale roughness.

P12, last paragraph: these observations are directly related to the governing equations and should not be presented as new results.

P13, l1-7: This paragraph and Figure 10 seem a bit out of place. Does this belong to the discussion of roughness? Maybe this should be moved to an earlier section.

P13, l17: It seems a bit exaggerated to conclude extensive irrigation of large amounts of green roofs are linked to 'better management of urban water cycles'

P13, l19: The combined impact of radiation trapping and shadows on the heating of

urban surfaces is not a new finding of this study.

P13, l24: Be careful, talking about effectiveness of changing morphometric conditions of the urban surface. The model setup used here is very simple and limited conclusions can be drawn on the effect of altered surface roughness of a real urban canopy.

- Check unites of roughness length in Table 2

- Include in Table caption of Table 3: 'For definition of symbols and units wee Table 1'

Technical comments

- check use of articles

- singular and plural forms often mixed up

---

## Author Comment (AC1) · 3 Apr 2016

Jiyun Song and Zhi-Hua Wang

zhwang@asu.edu

We would like to thank the reviewer for her/his constructive feedback and help in improving the quality of the manuscript. Below are our detailed responses to the comments. All changes are highlighted in the revised manuscript correspondingly. (The response and the revised manuscript with changes highlighted are also attached as supplementary files).

General comments:

Comment: This is not the first time model parameters are evaluated with such a stochastic approach. More insights should be provided on existing literature on this

area of research. The open questions and uncertainties should be listed and addressed specifically. It should be clearly stated why certain parameters are being tested.

Response: The stochastic approach adopted in this study, viz. the Subset Simulation is a versatile approach especially designed for small probability simulations (via an advanced importance sampling technique) The Subset Simulation approach have been used for risk, extreme events, and sensitivity analysis in a wide range of scientific and engineering areas, e.g. dynamics, seismology, fire safety, nuclear engineering, and geophysics, to name a few. The current study is one of its applications to urban climate study, and its first time application to coupled urban land-atmosphere model. We added discussion in the introduction part to put the current study in a bigger picture so readers can gain more insight into the Subset Simulation model and its application. In addition, we also phrased the specific research questions to be addressed by the current study in the introduction. The choice of certain parameters to be tested by the sensitivity study is justified in the context, which, in brief, is mainly based on the reported critical model parameters for physical modeling of land and atmospheric processes in the literature.

Comment: The conclusions drawn on the sets of model parameters are mostly to be expected and often directly obvious from the equations. The analysis should be rephrased so that these findings are not presented as if they were new results. Rather the benefits of the stochastic approach compared to other techniques should be emphasized. The conclusions sections should be rephrased accordingly.

Response: The novelty of our work lies in two aspects: (1) the focus on the atmospheric boundary layer (ABL) overlying a built environment (rather than in the urban canopy layer), and (2) the use of Subset Simulations for quantifying the sensitivity of modeling urban land-atmosphere interactions. We do not agree that the sensitivity of modeling parameters is obvious from the equations. For example, the canyon aspect ratio (building height to canyon width) representing the urban geometry, impact the urban surface energy balance in a very complicated and implicit way, through the action

of radiative view factors, leading to a nonlinear effect. This nonlinear effect has only been realized in a very recent study (Theeuwes et al., 2013, QJRMS) within the urban canopy level. In this study, we further identified the nonlinear effect as being able to "penetrate" into the ABL. In addition, any conclusive extension for quantification of model parameters and their sensitivity, from urban canopy to ABL, is never a trivial task. A good example is the set of hydrological parameters of urban vegetation (saturation, residual moisture content, and hydraulic conductivity) that was found to play a determinant role in regulating humidity and latent heat within urban canopies (Yang and Wang, 2014, Build. Env.) (rather "obvious", isn't it?), but do not have any significant effect in regulating the humidity level of the ABL), reason being that its effect is swept by the urban geometry. Last but not the least, we believe that being able to "intuitively" perceive the importance of model parameters from equations, is extremely important and requires deep understanding of the physics like the reviewer. Nevertheless, the importance of being able to quantify them using mathematical or statistical procedures (error analysis or sensitivity study) should not be downplayed, which on the one hand, confirms our physical intuition and the way the model is supposed to work, and on the other hand, often explores the less obvious part of and sheds mew lights on a modeling framework (e.g. the nonlinear effect of urban geometry). Nevertheless, we agree with the reviewer and thank his suggestion on emphasizing the stochastic approach. The conclusion part has been revised accordingly to reflect her/his suggestions.

Comment: It would also be appropriate to change the title to put the emphasize on the methodology rather than the parameter testing as this is the more relevant/new contribution of this work.

Response: The title has been changed to "Evaluating the impact of built environment characteristics on urban boundary layer dynamics using an advanced stochastic approach", to reflect the emphasis on methodology.

Specific comments:

Comment: P1, l12: Be careful with such statements: 'fraction of paved/vegetated terrains imposes more significant impact than the urban morphology'. When comparing different drivers for conditions in the atmospheric boundary layer, it is difficult to draw such generalized conclusions. In this study, extremely high fractions of irrigated green roofs are tested which will naturally have a big impact. It would have to be evaluated how much the morphology would have to change to achieve similar effects in order to make such a direct, generalized comparison.

Response: The statement has been rephrased in the revision to avoid drawing such generalized conclusions.

Comment: P1, l27: Make clear this is referring to the model setup: 'coupled' land-atmosphere processes?

Response: The term 'coupled land-atmospheric processes' is changed to 'land-atmosphere interactions' for better clarity.

Comment: P2, l9: should 'boundary characteristics' rather be 'boundary conditions'?

Response: We changed 'boundary characteristics' to 'boundary conditions'.

Comment: P2, l12: State which studies in the literature have used similar approaches to simulate the impact of varying input parameters, i.e. changing multiple parameters simultaneously. Which are the open research questions this study is aiming to address? Response: Exemplary applications of the Subset Simulations are added covering a rather wide range of fields including seismic risk, fire safety, spacecraft thermal control, and climatic extremes (Au et al., 2007; Thunnissen et al., 2007; Wang et al. 2011; Au and Wang, 2014; Yang and Wang, 2014). We also phrased the open research questions at the end of the introduction section, as "The sensitivity analysis in this study will therefore allow us to ask fundamental questions such as: How effective is a certain urban mitigation approach in modifying the CBL structure and to what elevation? What alternative strategies do we have in urban landscape planning in addition

to the popular options such as green/white roofs?"

Comment: P2, l15: define 'uncertain parameters'

Response: We added a definition of uncertain parameters as "parameters subject to variability and lack of deterministic values in the analysis of interest". We hope this helps to clarify this rather general and usually vaguely defined concept.

Comment: P2, l19: check grammar

Response: The sentence is rephrased for better clarity.

Comment: P2, l29: explain what is meant by 'sensitive quantification'

Response: The context in which this phrase is contained has been removed.

Comment: P3, l19: comment on advection, i.e. that it is not represented by the used setup

Response: The following discussion has been added on the advection term and the general surface energy balance closure problem, in the revised version. "Note that the thermal energy involved in advection, radiative flux divergence, and canyon air temperature variation is considered small when compared with the energy stored in urban surfaces (Nunez and Oke, 1977). It is noteworthy that in reality, the ideal surface energy balance condition is rarely observed whereas the energy imbalance is a norm rather than an exception, leading to a energy residual (see Foken, 2008 for a comprehensive review on this subject). In addition, a posteriori analysis of surface energy budgets found that only one percent of the residual variance can be attributed to advection and is not statistically significant (Higgins, 2012)."

Comment: P3, l21: what about the heat storage in the air volume within the street canyons?

Response: The heat storage in the canyon air volume is very small compared with the heat storage in soils and building envelops, given the heat capacity of air is only about

1/1000 of that of solids. The air storage term is therefore usually neglected.

Comment: P4, l4: where is the 'top of the surface layer'? mention blending height concept

Response: The surface layer of a CBL is not rigorously defined, but rather taken roughly as the lower 10% of the CBL (Stull, 1988), also referred to as "the constant flux layer". The overall CBL height is calculated via an analytical equation, viz. Equation (11) in this paper. The blending height specifically refers to the elevation where the spatial fluctuations due to horizontal heterogeneity are damped down to a small fraction of their magnitude at the surface (Phillip, 1997, BLM, 85-98). As numerical framework is a 1D column model and does not contain horizontal heterogeneity, the concept of blending height is not relevant. If the reviewer meant "mixing height" or "height of mixing layer", then this particular layer sits right on top of the surface layer and occupying the rest ∼90% of the total boundary layer. We have clarified this in the revision.

Comment: P4, l25: repetition with line 8

Response: The sentence is rephrased to avoid repetition.

Comment: P6, l12: Which database do the surface station data come from? How did you estimate the footprint of the radiosonde?

Response: The dataset for model evaluation was recorded by a network of wireless weather stations, deployed close to the radiosonde. Details of sensing instrumentation, data retrieval, and quality control can be found in Song and Wang (2015b, Build. Env. 94, 558-568) and are not repeated here. Footprint of the radiosonde was estimated using the analytical footprint model by Kormann and Meixner (2001, Bound.-Layer Meteorol., 99, 207-224). The footprint does not match the measurement area of the wireless sensor network exactly, but the latter is the closest (and best) dataset we can obtain for the model evaluation purpose.

Comment: P6, l16: To draw the conclusion that turbulent surface fluxes are also evaluated by the good agreement with the radiosonde profiles includes a series of assumptions. Also comment on the differences between observations and model profiles in the lowest layers where surface fluxes will have the most impact.

Response: The conclusion is removed to avoid ambiguity. For the difference between observations and model profiles in the lowest layers, this is mainly due to that the SCM in the modelling framework uses Monin-Obukhov similarity theory (MOST) for parameterizing the surface layer profiles. MOST assumes homogeneity of turbulence and surface conditions, which is rarely satisfied for the ABL over a built terrain. This clarification is added in the revision.

Comment: P7, l22: What exactly is meant by 'calibrated' input parameters? Are all roofs at mean roof height or is some variability assigned to the roof height?

Response: By "calibrated" parameters, we refer to the initial set of model parameters that were used in the model evaluation phase. These parameters are then allowed to vary (as "uncertain" parameters) in order to reflect possible landscape modification scenarios, e.g. the roof height takes a fixed value for model evaluation (Section 2), but is subject to variability in the sensitivity analysis (Section 3) for us to study the impact of urban geometry. For better clarity, the word "calibrated" is deleted from the context.

Comment: P7, l24: Why did you choose such high soil moisture levels?

Response: There are two main reasons: (1) the high soil moisture level is consistent with the management practice in our study area Phoenix, Arizona as a desert city where constant daily irrigation is applied for all urban vegetation (including green roofs, lawns, etc.) to maintain its eco-biological functionality; and (2) numerically, the study of green roofs is only meaningful when they remain "green". Prior study showed that a high initial moisture level best represents the model sensitivity of hydrothermal properties of green roofs, whereas a moderate to low initial moisture level causes green roofs to be out of function in later hours of a diurnal cycle, especially underestimate their

actual capacity of evapotranspiration (ET) and cooling effect due to high plant water stress and limited soil water availability during late afternoon hours.

Comment: P7, l27: Comment why these specific parameters were selected/ why the others are considered less critical.

Response: These specific parameters were selected based on our research objective in this study with the focus on the CBL dynamics impacted by landscape conditions. These parameters can be categorized into two main groups, representing the lower and upper boundary conditions, viz. the surface and the ABL top parameters respectively. More specifically, (1) for surface parameters: to compare the impact of green roof and conventional roofs, we selected parameters that could influence the surface energy partitioning above roof, including thermal parameters (albedo, heat capacity, thermal conductivity) and aerodynamic parameter (roughness length of momentum) for both roof types and hydraulic parameters (saturated water content, residual water content, saturated hydraulic conductivity) for green roofs. To investigate the impact of urban canyon geometry, we selected normalized roof width and aspect ratio (i.e. the ratio of roof height over road width) as two determinant parameters. And (2) for ABL top parameters: Two atmospheric parameters at ABL top, i.e. the entrainment rate and the virtual potential temperature gradient in the free atmosphere are selected, both parameters exert strong control on the evolution of the ABL dynamics. In addition, PDFs of these parameters are determined based on previous studies (Ouwersloot and Vilà-Guerau de Arellano, 2013; Wang et al., 2011a; Yang and Wang, 2014b) and local conditions in our study area. Care must be taken here that this particular selection of uncertain parameter space is by no means exhaustive or unique, and is subject to the limitation of parameterization used in the numerical framework and the subsequent analysis can, at best, represents only the model physics. This discussion is added to the revision.

Comment: P7, l30: How did you select physically realistic parameter space for uncertain parameters? Based on literature? Representing a specific type of city?

Response: The selection is based on both the literature and the prior study of the Phoenix metropolitan (references listed in the response above). The variability of the selected parameters in this study is therefore more representative of the cities in semi-arid or arid regions.

Comment: P7, l32: By 'dimensional parameters' you mean morphometric?

Response: By 'dimensional parameters' we specifically mean the parameters representing the geometry of the urban canyon. The term "geometric" is added for better clarity.

P8, l1: How did you set uniform distribution for entrainment rate and lapse rate?

Response: The uniform distribution for entrainment rate and lapse rate are set in the ranges [0.1, 0.3] m/s and [0.003, 0.007] K/m, respectively, based on Ouwersloot and Vilà-Guerau de Arellano (2013). We deem, to the best of our knowledge, that the variability of these parameters does not follow any particular probability distribution, but are most likely being equally probable in the prescribed ranges.

Comment: P8, l24: with 'log concavities' you are referring to the shape of the curves? Maybe this could be re-phrased to me something more descriptive.

Response: Log concavity refers to the shape of the curves of exceedance probability distribution, as well as the underlying statistical characteristics of this peculiar shape. This is a common phrase in probability theory, so we prefer to keep it as it is.

Comment: P9, l23: As Table 3 and Figure 7 have the same content, the table could be moved to supplementary material?

Response: Table 3 is removed from the revision to avoid the redundancy.

Comment: P9 l24: Normalised roof width not included in Table 1. Is a change in roof width connected to a change in plan area fraction?

Response: The parameter is added to Table 1. It is the geometric parameter commonly

used in canyon representation analogous to the planar area fraction (more commonly found in block representation).

Comment: P10, l2-13: move to introduction

Response: This sentence is moved to Introduction.

Comment: P10, L23: According to Table 1, aerodynamic resistance appears to be a set input parameter. Here you mention it to be a function of surface roughness. If this is the case this should be mentioned in the methods section.

Response: Table 1 only contains roughness length as input parameters. The aerodynamic resistance is not a set input parameter, but a function of roughness length based on Monin-Obukhov similarity theory (Mascart et al., 1995; Wang et al., 2013). This has been highlighted in section 4.4 of the revised manuscript.

Comment: P11, l29: Careful: most green roofs are constructed to be extensive, i.e. that they would not require irrigation but would use only natural water from precipitation. Comment on the representativeness of your model setup.

Response: In our study area, as well as in many cities in similar semi-arid and arid regions (e.g. Portland, Denver, etc.), green roofs require constant irrigation. As mentioned in response above, the current study is more representative of urban planning strategies in these cities.

Comment: P11, l32: this sentence is difficult to read. What is meant by 'detailed boundary layer physics : : : require further investigation'?

Response: The sentence is rephrased.

Comment: P12, l10: Are these results integrated over the whole day?

Response: These results are related to Fig. 9(b), where the vertical axes are labeled as 'critical' values, meaning that the maximum values of the 24-hours cycle are used, not integrated or averaged over the whole day.

Comment: P12, l19: Provide reference for the Zm – Zh relation

Response: Reference to Mascart et al. (1995, Bound.-Layer Meteorol. 72, 331-344) is provided.

Comment: P12, l23: Given resistances are directly related to turbulent fluxes, this is not a surprising result. Altering the roughness length is a fundamental change to urban morphology. Especially as roughness characteristics of individual facets combine to the local scale roughness.

Response: The roughness length we discussed here refers to the roughness above roof rather than the overall roughness of the entire built terrain. The distinction is made clear in Table 1 as roughness above roof or canyon, respectively. Therefore, different roughness length above roof is related to more "micro-level" management over rooftops, not the drastic change of urban morphology. For example, change from a paved roof (with roughness $\sim O(1 \text{ mm})$) to a vegetated roof (with roughness $\sim O(1 \text{ cm})$) will increase the roughness above roof easily by an order of magnitude; but this change has negligible effect on the overall roughness of the urban area (the latter is referred to as $z_{0,town}$ in Masson (2000) and is estimated as roughly 2/3 of the average building height).

Comment: P12, last paragraph: these observations are directly related to the governing equations and should not be presented as new results.

Response: The evolution of boundary layer height can be directly related to the governing equation, i.e. Eqn. (11), while the evolution of boundary layer temperature cannot. Thus we deem the discussion here, whether as new results to city practitioners or confirmation of the physical intuition of more experienced researchers, is appropriate.

Comment: P13, l1-7: This paragraph and Figure 10 seem a bit out of place. Does this belong to the discussion of roughness? Maybe this should be moved to an earlier section.

Response: This paragraph is moved to Section 2.3 where the stochastic modeling procedure is introduced.

Comment: P13, l17: It seems a bit exaggerated to conclude extensive irrigation of large amounts of green roofs are linked to 'better management of urban water cycles'

Response: The sentence is removed.

Comment: P13, l19: The combined impact of radiation trapping and shadows on the heating of urban surfaces is not a new finding of this study.

Response: In the conclusion part, we are not highlighting this counteracting effect of trapping and shading, but the nonlinear impact of urban morphology (viz. aspect ratio) on the convective boundary layer dynamics. See also our response to the general comments for relevant discussion. This part of concluding remarks is also revised for better clarity.

Comment: P13, l24: Be careful, talking about effectiveness of changing morphometric conditions of the urban surface. The model setup used here is very simple and limited conclusions can be drawn on the effect of altered surface roughness of a real urban canopy.

Response: We reiterate the limitation of the current study in the concluding remarks, so that readers will exercise their own discretion in interpreting the findings.

Comment: Check unites of roughness length in Table 2

Response: As stated above, the roughness length in this article only refers to the roughness above roof. So their values range 0.1-5 mm for paved roofs and 10-200 mm for vegetated surfaces.

Comment: Include in Table caption of Table 3: 'For definition of symbols and units see Table 1'

Response: Table 3 has been removed to avoid redundancy with Figure 8.

Technical comments

Comment: Check use of articles Singular and plural forms often mixed up

Response: The manuscript has been thoroughly checked and updated to ensure the correct use of articles and consistency in singular and plural forms (e.g. land-atmosphere interactions). We thank the reviewer for the meticulous examination of the manuscript.

Please also note the supplement to this comment:
http://www.atmos-chem-phys-discuss.net/acp-2015-955/acp-2015-955-AC1-supplement.zip
* * *

---

## Short Comment (SC1) · 4 Apr 2016

We would like to thank the reviewer's comments and the constructive feedback during the interactive discussion in improving the quality of the manuscript. The reviewer pointed out the significance of the current study and promising possibilities in extending the results to various future research directions, which is highly appreciated.
* * *